# CIDA-3D: CONFORMAL INFERENCE AIDED UNSUPERVISED DOMAIN ADAPTATION FOR 3D-AWARE CLASSIFICATION

## ABSTRACT

Cognitive Science studies show that human perception becomes robust to occlusions and other nuisances due to internal 3D representations of objects. This idea has been incorporated into computer vision models to improve their ability to understand and reason about the 3D world. However, collecting 3D annotations in vision datasets is expensive. This makes the robustness of the perception model to distribution shifts challenging. We introduce Conformal Inference aided unsupervised Domain Adaptation (CIDA)-3D for the complex setting of multiclass pose estimation. Our method adapts category level pose estimation (3D) models in nuisance ridden target domains directly from images without class label information, by harnessing uncertainty in model predictions (using conformal sets). This allows for significantly better and computationally efficient adaptation to target domains with synthetic and real-world noise. We also show a robust adaptation from fully synthetic data to complex real-world domains. To the best of our knowledge, this method is the first to attempt unsupervised domain adaptation for robust 3D-aware classification and multiclass pose estimation in real-world scenarios by adapting models trained on procedurally generated synthetic data.

## 1 INTRODUCTION

Remarkable progress has been observed in recent years in the area of 3D object representation learningJesslen et al. (2023), revolutionizing applications ranging from robotics Du et al. (2019); Wang et al. (2019a); Wong et al. (2017); Zeng et al. (2017) and augmented reality Marchand et al. (2016); Marder-Eppstein (2016); Runz et al. (2018), etc. Cognitive science studies (Neisser, 2014; Yuille & Kersten, 2006) have often theorized that robustness to OOD inputs, occlusions, and other nuisances is often due to implicit 3D object representations built into visual processing of humans and similar mammals. Several works Jesslen et al. (2023); Wang et al. (2021a); Yang et al. (2023); Wang et al. (2023); Stark et al. (2010); Choy et al. (2015); Zeeshan Zia et al. (2013) have utilized similar hypotheses to build robust 3D object representations for different computer vision tasks such as 3D object pose estimation, shape identification, robust image classification, etc. Most previous works utilizing object 3D pose information are focused on the problem of 3D or 6D pose estimation. Instance-level He et al. (2021; 2020); Park et al. (2019); Peng et al. (2019); Tremblay et al. (2018); Wang et al. (2019a); Xiang et al. (2018) pose estimation is most common and requires instance-specific 3D data and priors. Category-level methods Chen et al. (2020); Chen & Dou (2021); Lin et al. (2021); Tian et al. (2020); Wang et al. (2019b; 2021b) are more efficient but still require 3D information, e.g. object depth map Wang et al. (2019b); Lin et al. (2021); Lee et al. (2022) or point clouds Lee et al. (2023). Extensions to multiclass pose estimation, which is often a prerequisite for problems such as *3D aware classification* are even rarer. We define **3D Aware Classification** as the problem of image object classification where the model prediction is conditioned on implicit or explicit 3D representation of the object in the image.

Recent works Wang et al. (2023); Jesslen et al. (2023) have shown that 3D-Aware classification is a robust alternative to conventional 2D-only image classification. However, it has not been clear how to extend these methods beyond strictly supervised settings on relatively simpler datasets. This is because, unlike image data, which are widely available, real-world 3D data is scarce, restricting the development of 3D-aware models. To remedy this, our work focuses on the problem of unsupervised

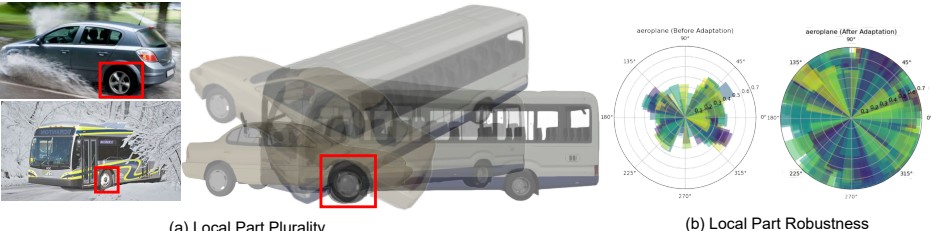

(a) Local Part Plurality        (b) Local Part Robustness

Figure 1: Our method utilizes following key observations - (a) **Local Part Plurality**, i.e. the inherent object identification ambiguity that occurs when we can only see a part of the object since similar parts may occur in different objects and in different poses. We utilize this ambiguity to update the local vertex features across different categories which roughly correspond to object parts, even when the object in the image is different. (b) **Local Part Robustness** As explained in Kaushik et al. (2024), refers to the fact that certain parts (e.g., headlights, wheels in a car) are less affected in OOD data. This has been verified in Kaushik et al. (2024), and we find similar evidence in our multiclass setting. The figure represents the percentage of robustly detected vertex features on average per image in a target domain(OOD-CVZhao et al. (2023)) for airplane category *before (left)* and *after (right) adaptation*. Similar to Kaushik et al. (2024), we find that few vertices are detected robustly even before adaptation which our method leverages in the multi-class setting.

domain adaptation (UDA) for 3D-Aware Classification and multiclass pose estimation. We design a model that is capable of adapting to a real-world target domain in an unsupervised manner **without requiring any kind of 3D data or object labels and using only unlabeled images in the target domain**.

Previous works Lee et al. (2022; 2023) have largely focused on only semi-supervised category-level pose estimation and still require some 3D information. A recent seminal work Kaushik et al. (2024) has succeeded in image-only *unsupervised* domain adaptation for estimating 3D poses at the category level. They utilized the idea that certain parts of an object exhibit invariance in out-of-distribution scenarios. In this paper, we extend this idea to a multi-category setting. We find that different parts features of a target domain image may be utilized to update parts of neural mesh models of different object categories despite noisy pose estimation (Figure 1).

Like Kaushik et al. (2024), our source model is based on neural mesh models (Kortylewski et al., 2020; Wang et al., 2021a; 2023; Ma et al., 2022; Jesslen et al., 2023) used for supervised 3D/6D object pose estimation and 3D-Aware image classification Jesslen et al. (2023). These methods represent objects as cuboid meshes and learn neural activations at each vertex, enabling pose estimation through feature-level rendering and optimization. Kaushik et al. (2024) enabled unsupervised domain adaptation (UDA) for 3D pose estimation by updating cuboid mesh features to estimate robust subcomponents of objects. We extend this to a multi-category UDA setup. Our method, CIDA-3D, updates a model of multiple cuboid meshes and a single neural backbone to classify and estimate the 3D pose of unlabeled target domain objects. We present experimental results showing how CIDA-3D adapts from synthetic to complex real-world target domains. Our method learns from synthetic data alone, enabling the use of 3D knowledge in computer vision without real-world 3D ground-truth data.

In summary, we make several important contributions in this paper.

1. We introduce CIDA-3D, the first method known to do **image only unsupervised domain adaptation for 3D-Aware Classification and multiclass 3D pose estimation**.

2. CIDA-3D builds on 3DUDA(Kaushik et al., 2024) and uses local part plurality and robustness (Figure 1) to adapt to nuisance-ridden domains with unlabeled images.

3. We utilize weighted Conformal Prediction for covariate shift Tibshirani et al. (2019), achieving confident prediction sets that minimize computational overhead and divergence issues of naive adaptation.

4. We evaluate our model on real-world nuisances such as shape, texture, occlusion, and image corruptions, demonstrating robust adaptation. CIDA-3D allows adaptation from a synthetic source domain to a nuisance-filled real-world target domain.

## 2 RELATED WORK

*Neural Mesh Models* It refers to a family of neural modelsWang et al. (2021a; 2023); Jesslen et al. (2023); Ma et al. (2022) that learn a 3D pose-conditioned model of neural features and predict

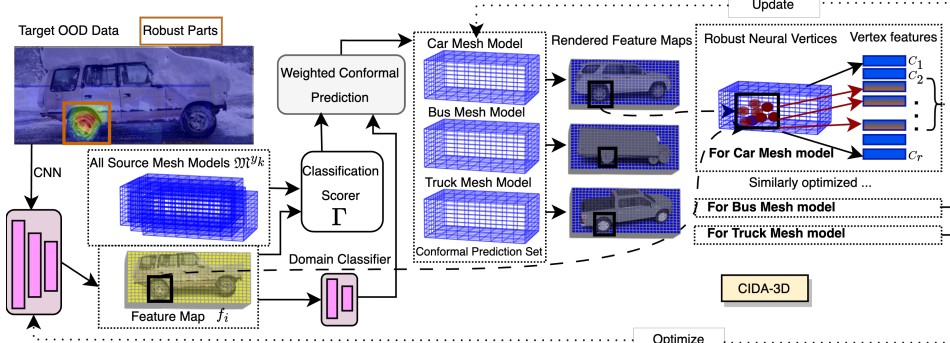

**Figure 2:** We extract neural features from CNN backbone $f_i = \Phi_w(\mathcal{X}_\mathcal{T})$ and use them along with all source neural mesh models ($\mathfrak{M}^{y_k}$) to get the classification scores ($\Gamma$) as described in Jesslen et al. (2023). To perform adaptation, a domain classifier trained to distinguish source features from target features is used for weighted conformal prediction, giving a prediction set ($S_i$) of classes to which a target image confidently belongs. Feature maps are rendered from the source mesh models of these classes (using vertex features $C_r$) and the pose estimate is optimized using render-and-compare. For this incorrectly estimated global pose, we measure the similarity of each individual visible vertex feature with the corresponding image feature vector in $f_i$ *independently* and update individual vertex features using average feature vector values. All predicted mesh models are then updated using these changed vertices and the backbone is optimized using the neural mesh. More details in section 3.

3D pose by minimizing the reconstruction error between the actual and rendered feature maps. This optimization approach helps circumvent the intricate loss landscapes that can emerge from performing pixel-level render-and-compare. For example, Wang et al. (2019b) predicted the pose of the object by solving a rigid transformation between the 3D model (M) and the NOCS mapsWang et al. (2019b) using the Umeyama algorithmPavlakos et al. (2017). While Iwase et al. (2021) used differentiable Levenberg-Marquardt optimization for feature learning, Wang et al. (2021a) and Ma et al. (2022) learned contrastive features for the 3D model (M) within a similar render-and-compare framework.

*Domain Adaptation for 3D Pose Estimation* Several semi-supervised approaches exist, such as those described in Fu & Wang (2022); Peng et al. (2022), which often necessitate labeled target-domain images and 3D data. Even methods like Lee et al. (2022; 2023) require instance depth data, point clouds, or segmentation labels during inference. Alternatively, methods such as Yang et al. (2023) generate synthetic data and combine them with a limited amount of real annotated data for synthetic-to-real semi-supervised domain adaptation. To the best of our knowledge, only one recent workKaushik et al. (2024) other than ours is capable of doing image-only object 3D pose estimation. However, even Kaushik et al. (2024) cannot do UDA for multi-class pose estimation.

*Conformal Inference* We utilize concepts from some seminal prior works Tibshirani et al. (2019); Shafer & Vovk (2008); Lei et al. (2018); Park et al. (2020) which show theoretical guarantees of high confidence conformal predictions under i.i.d. as well as covariate shift settings. These effective uncertainty handling techniques have only recently started getting traction Yang & Pavone (2023); Sankaranarayanan et al.; Belhasin et al. (2023).

## 3 METHODOLOGY

Similarly to 3DUDA Kaushik et al. (2024), we build on neural mesh models Wang et al. (2021a; 2023); Ma et al. (2022). Our source model uses a similar method to that in a concurrent work(Jesslen et al., 2023). This model performs 3D-Aware Classification and pose estimation but cannot be easily adapted to a target domain using classification pseudo-labels since it is not directly supervised for classification. Unlike Jesslen et al. (2023), our pose estimation depends on the class predicted by the classification inference, while (Jesslen et al., 2023) handles these tasks independently. The following section briefly introduces this source model. Figure 2 provides a visual explanation. Refer to Jesslen et al. (2023) or our appendix for further details.

**Notation** We follow the notation introduced in Kaushik et al. (2024). We define a set of object categories $Y = \{y_0, y_1, ...y_k\}$ where $|Y|$ is the total number of categories. We define three sets of parameters: a CNN backbone $\Phi_w$ that is used as a feature extractor, a clutter model $\mathcal{B}$ of background features, and neural cuboid mesh $\mathfrak{M}^{y_k}$ for each object category $y_k$. We denote the neural feature representation of an input image $\mathcal{X}$ as $\Phi_w(\mathcal{X}) = F^a \in \mathcal{R}^{H \times W \times d}$. Where $a$ is the output of layer $a$

of a deep convolutional neural network backbone $\Phi_w$, with $d$ being the number of channels in layer $a$. $f_i^a \in \mathcal{R}^d$ is a feature vector in $F^a$ at position $i$ on the 2D lattice $P$ of the feature map. We drop the superscript $a$ in subsequent sections for notational simplicity.

## 3.1 SOURCE MODEL: 3D OBJECT REPRESENTATION LEARNING

Our source model learning process is similar to a concurrent workJesslen et al. (2023) that builds on previous worksMa et al. (2022); Wang et al. (2021a) performing pose estimation by learning neural mesh models conditioned on 3D object poses and estimating pose using feature-level render and compare.

The neural mesh model aims to capture the 3D information of the foreground objects. For each object category $y_k$, the source model defines a neural mesh $\mathfrak{M}^{y_k}$ as $\{\mathcal{V}, \mathcal{C}\}$, where $\mathcal{V}_y = \{V_r \in \mathbb{R}^3\}_{r=1}^R$ is the set of vertices of the mesh and $\mathcal{C}_y = \{C_r \in \mathbb{R}^c\}_{r=1}^R$ is the set of learnable neural features. $r$ denotes the index of the vertices. $R$ is the total number of vertices per class. We also define a clutter model $\mathcal{B} = \{\beta_n\}_{n=1}^N$ to describe the backgrounds that are shared amongst all classes. $N$ is a prefixed hyperparameter. For a given object pose or camera viewpoint $g$, we can render the neural mesh model $\mathfrak{M}^{y_k}$ (denoted simply by $\mathfrak{M}$ below for simplicity) into a feature map using (differentiable) rasterization Kato et al. (2020). We can compute the object likelihood of a target feature map $F \in \mathcal{R}^{H \times W \times D}$ as

$$p(F|\mathfrak{M}, g, \mathcal{B}, y) = \prod_{i \in \mathcal{FG}} p(f_i|\mathfrak{M}, g, y) \prod_{i' \in \mathcal{BG}} p(f_{i'}|B), \tag{1}$$

where $\mathcal{FG}$ and $\mathcal{BG}$ denote the foreground and background pixels, respectively. $\mathcal{FG}$ is set of all the positions in the 2D lattice $P$ covered by the mesh $\mathfrak{M}$ and $\mathcal{BG}$ are the positions that are not. We define $P(f_i|\mathfrak{M}(V_r, C_r), g) = Z[\kappa_r] \exp\left(\kappa_r f_{i \to r}.C_r\right)$ as a von Mises Fisher (vMF) distribution with mean $C_r$, concentration parameter $\kappa_r$ and normalization constant $Z$. For computational simplification, we fix $\kappa$ which reduces $Z[\kappa]$ to a constant value as well. The basic idea is to learn cuboid neural mesh features conditioned on 3D pose of an object and maximize the dot product $f_i.C_r$ where the image features are obtained from a single neural feature extractor and the neural mesh features belong to the respective image category.

We utilize contrastive learning to learn the cuboid neural mesh features. The formulation is as follows where $\mathcal{N}_r$ denotes the vertices near $r$ i.e. the neighborhood of the vertex $r$ and $y$ is the category of the image. During training, the ground truth pose specifies the image feature-vertex feature correspondence (denoted $f_{i \to r}$). $R$ defines set of all visible vertices. We maximize the probability that an image feature is generated by the correct mesh vertex feature within a class as well as among all other classes and background features:

$$\frac{P(f_{i \to r}|C_r)}{\sum_{l \notin \mathcal{N}_r, l \in R}^{C_l \in \mathfrak{M}^y} P(f_{i \to l}|C_l) + \sum_{n=1}^N P(f_{i \to n}|\beta_n) + \sum_{m \in R, m \notin \mathcal{N}_r}^{C_m \notin \mathfrak{M}^y} P(f_{i \to m}|C_m)} \tag{2}$$

Vertex features are updated using simple momentum updates, and background features are learned by randomly sampling background features from new training batches using the First-In last-Out approach Jesslen et al. (2023).

## 3.2 INFERENCE FOR 3D AWARE CLASSIFICATION

**Pose Estimation using Render-and-Compare** Feature-level render-and-compare is used for estimating 3D object pose. We can infer the 3D pose $g$ of the object $y$ by minimizing the negative log likelihood of the model. Specifically, we first extract the neural features of the image $F = \Phi_w(\mathcal{X})$ from the CNN backbone. We define an initial pose $g_{init}$ using random initialization or by pre-rendering and matching some random poses. Using the initial pose, we render the neural mesh $\mathfrak{M}$ into a feature map $F' \in \mathcal{R}^{H \times W \times D}$. The projected feature map is divided into $\mathcal{FG}$ and $\mathcal{BG}$, depending on which pixels on the feature map are covered by the projected mesh features. We compare the rendered feature map and the image feature map position-wise. Given that the feature vectors are normalized and considering a constant $\kappa$, the loss can be refactored as a simple reconstruction loss. The pose $g_{init}$ is optimized by minimizing following using stochastic gradient descent:

$$\mathcal{L}_{rec} = 1 - \ln p(F|\mathfrak{M}, g, \mathcal{B}) = 1 - (\sum_{i \in \mathcal{FG}} f_i * f_i' + \sum_{j \in \mathcal{BG}} f_j * \beta). \tag{3}$$

**Classification Using Geometry-Independent Feature Matching**    A trivial way to classify images for these neural mesh models is to perform a render-and-compare-based gradient pose optimization for every category and compare the final reconstruction loss. However, this is a computationally expensive approach, which becomes untenable when we have a significant number of categories to choose from. To remedy this, a geometry-independent inference method is proposedJesslen et al. (2023). The foreground likelihood and the background likelihood are calculated at every position in the feature map using all vertex features (across all categories) and background clutter features. For all positions where foreground likelihood exceeds background likelihood, the maximum values are summed depending on which category the maximizing (most similar to image feature) vertex belongs to. These values are then normalized and compared for the final prediction. This is similar to conventional CNNs, where we can construe the neural vertex features and background features individually as one-dimensional convolutional kernels. The final prediction of this inference method can be formulated as follows:

$$\Gamma(\mathcal{X}_k) = \sum_{f_i \in F} \max\{\max_{C_r \in \mathfrak{M}^{y_k}} f_i \cdot C_r, \max_{\beta_n \in \mathcal{B}} f_i \cdot \beta_n\}; \quad \hat{y} = \arg\max_k(\Gamma(\mathcal{X}_k)) \tag{4}$$

$$\tag{5}$$

In an IID scenario, we find that coherent vertex-feature correspondence found using differentiable render-and-compare (Equation 3) is retained even when we utilize aforementioned geometry independent feature matching for inference (Equation 4). This means that the vertex features that minimize the reconstruction error during pose estimation (using render and compare) are largely those that are activated maximally during independent feature matchingJesslen et al. (2023).

However, this is no longer true in an out-of-distribution scenario. Predictions of classification inference and pose estimation often diverge. An example of this is provided in the Ablation Section in our appendix.

### 3.3    CIDA-3D: UNSUPERVISED DOMAIN ADAPTATION FOR 3D-AWARE CLASSIFICATION AND MULTI-CLASS POSE ESTIMATION

As the predictions from our fast, unconstrained model diverge from the slow, 3D-constrained render-and-compare estimates in an OOD scenario, inference becomes uncertain. Running render-and-compare for all objects and samples to verify fast predictions is computationally impractical. We cannot update our model using classification pseudo-labeling methods due to the lack of direct classification loss supervision. Methods like Kaushik et al. (2024) require knowing the ground truth class for updating the neural mesh model. Establishing if Kaushik et al. (2024)'s hypothesis on local part robustness and ambiguity applies to a multiclass setting is also challenging. Our method, CIDA-3D, addresses these issues by using uncertainty quantification from Conformal Prediction and extending Kaushik et al. (2024)'s hypothesis on Local Part Robustness (as described in Figure 1) to a multiclass setting, as explained in Figure 1.

**Using Local Part Robustness**    To adapt to an OOD target domain, we use the concept of local part robustness, as shown in Figure 1. Kaushik et al. (2024) showed that local part robustness can be exploited to update neural mesh models ($\mathfrak{M}^{yk}$) and the CNN backbone ($\Phi_w$) for single class pose estimation. We show that we can use the same intuition to adapt these models to perform 3D-Aware classification on target domain data. This is possible due to what we refer to as *Local Part Plurality* hypothesis (Figure 1). In layman terms, it refers to the inherent object identification ambiguity that occurs when we can only see a part of the object, since similar parts may occur in different objects and in different poses. We utilize this ambiguity (in terms of neural mesh vertex features) to update the local vertex features across different categories which roughly correspond to object parts, even when the object in the image is different. In addition, we also establish that the local part robustness hypothesis also stands in a multi-class setting (Figure 1 and ablation Section) and there are individual robust neural mesh vertices which remain unchanged or fewer changes across domains. Note that in Kaushik et al. (2024), adaptation was achieved in a category-level pose estimation task (where the class $y_k$ of the object was already known), which is a simpler problem with ground-truth knowledge of which mesh model needs to be updated.

As described in subsection 3.2, the 3D-aware classification scores for each class ($\Gamma$) can be calculated using Equation 4 Jesslen et al. (2023). For our classification task, we do not have access to

the target data labels. One naive way to achieve adaptation in this harder case is by treating the top prediction as a pseudolabel and updating the corresponding mesh model (using the locally robust method in Kaushik et al. (2024)). As the source model does not work well in the target domain, the top prediction is often wrong, and this approach creates a problem with noisy updates. This is analogous to using noisy pseudo-global updates (with potentially large pose error) instead of robust local updates to perform adaptation in 3D pose estimation, which has been shown to be problematic in Kaushik et al. (2024). In fact, after testing this approach on Corrupted-Pascal3D+, we found that the source models adapt very slowly and insufficiently (details can be found in our Ablation section).

Another way to adapt is by updating all mesh models ($\mathfrak{M}^{y_k}$) with locally robust parts. This method is computationally prohibitive as it requires render-and-compare for each model. Furthermore, it produces irrelevant updates in unrelated classes, impairing pose estimation. In an OOD-CV (shape) experiment, we found that while classification accuracy increased slightly, pose estimation accuracy dropped significantly and the process was much slower (details in Ablation Section).

To address these problems, we propose using conformal prediction Tibshirani et al. (2019) to obtain a set of predicted objects that contains the true class with high probability. This approach avoids both slow and divergent adaptation issues.

**Conformal Prediction**     Given a calibration set $D_C = \{\mathcal{X}_j, y_j\}_{j=1}^N$ of $N$ input and target (class) $\in$ $\mathcal{Y}$ pairs, drawn i.i.d. from an unknown distribution, conformal prediction provides a set predictions $f(\mathcal{X}_{j+1}) = S_{j+1} \subset \mathcal{Y}$ for a new sample $\mathcal{X}_{j+1}$ satisfying *exchangeability* (distribution is invariant of the order in which the points are presented Lei et al. (2018)) such that the true class of this sample, $y_{j+1} \in S_{j+1}$ with high probability (parameterized by $\alpha$). More specifically, $P(y_{j+1} \in S_{j+1}) \geq 1 - \alpha$. To give this conformal prediction guarantee, a non-conformity score $\mathcal{S}_f(\mathcal{X}_j, y_j)$ measures how well a new sample $(\mathcal{X}_j, y_j)$ conforms to the training set which is used to learn a predictor $f$. This can be as simple as disagreement between the prediction and true target, i.e. $\mathcal{S}_f(\mathcal{X}_j, y_j) = 1 - f(\mathcal{X}_j)^{y_j}$ where $f(\mathcal{X}_j)^{y_j}$ denotes the classification score assigned by $f$ on class $y_j$. The non-conformity scores are calculated for all samples in the calibration set ($D_C$), sorted and $1 - \alpha$ quantiles are calculated. The final output for a new sample $\mathcal{X}_{j+1}$ is a set of classes $S_{j+1}$ such that the non-conformity score of this sample is upper bounded by the quantile.

**Tackling exchangeability**     Notice that exchangeability is a strong requirement for these conformal prediction guarantees to hold. However, as we work in an unsupervised adaptation setting, the calibration set (required to give such guarantees) is not from the target domain. The exchangeability conditions are violated because the target domain has a different data distribution (a standard assumption of covariate shift where the marginal distribution $P(X)$ of image features changes between the source and target domains, but the conditional distribution $P(Y|X)$ remains the same). To address this problem, we use conformal prediction under covariate shift Tibshirani et al. (2019) by weighting the nonconformity scores of each sample in the calibration set with a likelihood ratio $P_T(X)/P_S(X)$.

In practice, it is difficult to estimate marginal densities $P_T(X)$ and $P_S(X)$. Instead, we fit a domain classifier on features extracted from the CNN back-end using images from the source domain ($\Phi_w^S(\mathcal{X})$) and target ($\Phi_w^T(\mathcal{X})$) domains. This classifier gives a score to each sample which we use as a proxy for the likelihood ratio $P_T(X)/P_S(X)$ to weight our calibration set. Note that this works best when there is some support overlap of image features between the source and target domains. The calibration set *looks* exchangeable with respect to the target distribution and makes the prediction set conform better to it.

The following steps describe our whole adaptation method:

1. Train a domain classifier to distinguish image features $\Phi_w(\mathcal{X})$ of source and target domains.

2. Use domain classification scores as a proxy for $P_T(X)/P_S(X)$ (importance weights). A weighted calibration set is used to perform conformal prediction for target samples, i.e. we get a prediction set $S_i$ for each target input $\mathcal{X}_i^T$.

3. 3DUDAM: Following Kaushik et al. (2024), obtain CNN features ($f_i$) for target images from the backend $\Phi_w$ and use predicted class mesh models (from prediction sets obtained in previous step) to generate rendered neural vertex features $C_r$. The robustness of a vertex

---

**Algorithm 1** Domain Adaptation for 3D Aware Classification(**CIDA-3D**)

---

**Input:** Source data $D_S = \{(\mathcal{X}_i^S, y_i^S)\} \sim P_S(X, Y)$, Calibration data $D_C = \{(\mathcal{X}_i^C, y_i^C)\} \sim P_S(X, Y)$, Target data $D_T = \{(\mathcal{X}_i^T) \sim P_T(X)\}$, source models $\mathfrak{M}^{y_k}$, source CNN backend $\Phi_w$ and classification scorer $\Gamma$.

**for** time step $t = 0, 1, ...$ until convergence **do**
    Domain Classifier $\Psi^t \leftarrow$ Trained using $\Phi_w^t(\mathcal{X}_i^S)$ and $\Phi_w^t(\mathcal{X}_i^T)$.          $\triangleright \Phi_w^0 = \Phi_w$
    Calibration weights $W_i \leftarrow \Psi^t(\mathcal{X}_i^C)$
    Prediction set $S_i \leftarrow \mathrm{CP}(\Gamma, W_i, \mathcal{X}_i^T)$         $\triangleright$ Conformal Prediction for target samples.
    **for** each target image $\mathcal{X}_i^T$ **do**
        **for** $y_k \in S_i$ **do**
            $\mathfrak{M}_{t+1}^{y_k} \leftarrow 3\mathrm{DUDAM}(\mathfrak{M}_t^{y_k}, \mathcal{X}_i^T, \Phi_w^t)$         $\triangleright$ Update predicted mesh models. (3)
        **end for**
        **for** $y_k \notin S_i$ **do**
            $\mathfrak{M}_{t+1}^{y_k} \leftarrow \mathfrak{M}_t^{y_k}$         $\triangleright$ Keep other mesh models the same.
        **end for**
        $\Phi_w^{t+1} \leftarrow 3\mathrm{DUDAC}(\Phi_w^t, S_i)$         $\triangleright$ Update CNN backend. (Equation 6)

---

    feature is calculated using the similarity
    $\mathcal{L}_{sim}(f_{i \to r}, C_r) = Z[\kappa_r] \exp\left(\kappa_r f_{i \to r}^T C_r\right)$ thresholded by $\delta_r$.
    Local robust vertex features of all mesh models (in the conformal prediction set) are updated using
    $C_r^{t+1} \leftarrow (1 - \tau)C_r^t + \tau \frac{1}{n}\sum_n f_{i \to r}, \quad \forall f_i \ni \mathcal{L}_{sim}(C_r, f_{i \to r}) > \delta_r$. Here, $\tau$ is the momentum hyperparameter. Trivially, we can set it to $0.5$. However, in this work, we empirically find that using $\mathcal{L}_{rec}$ to set the value of $\tau$ gives better results. We define $\tau = \max(0.8 * (1 - \mathcal{L}_{rec}), 0.1)$

4. 3DUDAC: Similar to Kaushik et al. (2024), update the CNN backend but using the predicted set of classes and a corresponding loss function as described in Equation 6 which can be derived from Equation 2.

We update our CNN backbone by optimizing the following loss function:

$$\mathcal{L} = -\zeta \sum_{r \in R_v} \log \frac{e^{\kappa f_{i \to r} C_r}}{\sum_{l \in R, l \notin \mathcal{N}_r}^{C_l \in \mathfrak{M}^y} e^{\kappa f_{i \to l} C_l} + \sum_{n=1}^N e^{\kappa f_{i \to n} \beta_n} + \sum_{m \in R, m \notin \mathcal{N}_r}^{C_m \notin \mathfrak{M}^y} e^{\kappa f_{i \to m} C_m}}, \quad (6)$$

where $R_v$ denotes all visible vertices for the input image $\mathcal{X}$. $\mathcal{N}_r$ denotes the vertices near $r$. In practice, we define a parameter $\zeta$ that is a weighting parameter that is $1 - \mathcal{L}_{rec}$ if the size of the prediction set is $> 1$. Subsequently, the estimated pose $g'$ is recalculated with the updated neural mesh models, and the CNN backbone is updated by gradient descent iteratively with the Equation 6. We iteratively update subsets of vertex features, recalculate the conformal prediction sets and finetune the CNN backbone till convergence in an EM type manner. In practice, to avoid false positives and encourage better convergence, we establish a few conditions in our selective vertex feature adaptation process. We fix a hyperparameter $\psi_n$ that controls the least number of local vertices detected to be similar ($5 - 10\%$ of visible vertices). We also drop samples with low global similarity values ($L_{rec} \geq 0.4$) during the backbone and vertex update. To save computational overhead, we can fix $\kappa$ for the loss calculation.

## 4 EXPERIMENTS

**Setup** We follow a conventional unsupervised domain adaptation setup Hoyer et al. (2023); Jin et al. (2019); Zhang et al. (2019). During adaptation and inference, only RGB images from a target domain set are provided to the model, trained in a supervised manner on source domain data. Unlike previous works, no 3D information, depth data, or point cloud from the target domain is provided. Contrary to Kaushik et al. (2024), we do not provide a category label for the target domain images. The model predicts the category and estimates the 3D pose of the object. Ensemble methods are not considered in this work.

*Benchmarks* Methods are evaluated on three benchmarks. The source model is trained on IID samples and adapted to OOD data with individual and combined nuisances. The first benchmark, OOD-CVZhao et al. (2023), includes real-world nuisances like context and weather for 10 categories. The second benchmark involves domain adaptation from real sources to synthetically corrupted targets. In Imagenet-CHendrycks & Dietterich (2019), Pascal3D+Xiang et al. (2014) (Table2), data are corrupted with noises like shot noise and fog from Imagenet-C. The third benchmark evaluates adaptation from synthetic to real-world nuisance-ridden domains. This UDA benchmark trains on synthetic data and adapts to real-world nuisances. Using Yang et al. (2024); Ma et al. (2023), synthetic images and 3D poses for 5 object categories are generated. Models are then adapted and evaluated on OOD-CVZhao et al. (2023) data. This shows domain adaptation methods like CIDA-3D help models learn 3D knowledge from noisy real-world images, applicable to other computer vision tasks.

*Evaluation* For Classification, we use prediction accuracy as a metric. For 3D pose estimation, we aim to recover the 3D rotation parameterized by azimuth, elevation, and in-plane rotation of the viewing camera. We follow previous works like Zhou et al. (2018); Kaushik et al. (2024); Ma et al. (2022) and evaluate the error between the predicted rotation matrix and the ground-truth rotation matrix: $\Delta(R_{pred}, R_{gt}) = \frac{||logm(R_{pred}^T R_{gt})||_{\mathcal{F}}}{\sqrt{2}}$. We report the accuracy of the pose estimation under common thresholds, $\frac{\pi}{6}$ and $\frac{\pi}{18}$.

**Baseline Models** In addition to the comparison with other 3D-Aware Classification methodsWang et al. (2023); Jesslen et al. (2023), we also compare with classification only and pose estimation only methods. Since our work is the first to attempt to solve 3D-Aware UDA problem, we compare our results to common classification-only UDA methods Cui et al. (2020); Jin et al. (2019); Zhang et al. (2019); Long et al. (2018); Na et al. (2021); Hoyer et al. (2023); Wei et al. (2021); Liu et al. (2021); Mirza et al. (2022); Liang et al. (2022); Rusak et al. (2021); Schneider et al. (2020) which have been shown to be the state-of-the-art on various classification-only robustness datasets.

**Implementation Details** An Imagenet pretrained Resnet50 is used as a common feature extractor for our source model. The cuboid mesh is defined for each category with features obtained from the common backbone. The source model is trained for 800 epochs with a batch size of 32 using an Adam optimizer in a fully supervised manner. Similarly to Jesslen et al. (2023); Wang et al. (2021a), during inference (for pose estimation), 144 poses are pre-rendered into features from the neural meshes and the one with the lowest reconstruction loss is chosen as the initial pose which is then optimized using gradient descent. For every adaptation step, we require a minimum batch size of 32 images for selective vertex and feature extractor updates. We choose a classification prediction set of 3 or fewer samples and perform pose estimation for these predictions. Samples with very low global reconstruction similarity ($< 0.4$) are removed from the update, and samples with very high global similarity ($> 0.85$) are fully used for vertex feature updates. Inference takes 0.21 seconds per sample on an RTX 3090. Our adaptation model is implemented in PyTorch (with PyTorch3D for differential rasterization) and takes around 4 hours to train on 2 A5000 GPUs.

### 4.1 RESULTS AND ANALYSIS

**OOD-CV** Table 1 shows Unsupervised Domain Adaptation results for Classification and multi-class pose estimation on OOD-CV Zhao et al. (2023), containing real-world images with nuisances like pose, texture, context, and weather. Our results, compared to SOTA UDA methods, validate that our method leverages 3D knowledge to enhance model robustness against real-world OOD nuisances. Even our source modelJesslen et al. (2023) outperforms many classification-only domain adaptation methods, highlighting the importance of 3D knowledge. Our method significantly outperforms all models and bridges the domain gap.

**Pascal3D→Corrrupted-Occluded-Pascal3D+** Table Table 2 shows results for UDA in Classification and multi-class 3D pose estimation. Synthetic corruption of level 5 from Imagenet-CHendrycks & Dietterich (2019) is applied to the validation dataset representing the target domain. The benchmark includes 3 levels of occlusion (0%, F1G1 - 20 − 40% occlusion in both foreground and background, and F2G2 - 40−60% occlusion) in addition to the corruptions, making it a complex setup. Occluded images from Occluded-Pascal3d+ datasetWang et al. (2020) are not shown to mod-

Table 1: Unsupervised Domain Adaptation for Classification and Multi-Class 3D Pose Estimation on OOD-CV 67 dataset (Metrics: $Acc.$: Classification Accuracy, $\frac{\pi}{6}Acc.$: 3D pose estimation accuracy; higher is better)

| | Acc.⬆ | $\frac{\pi}{6}$Acc.⬆ | Acc.⬆ | $\frac{\pi}{6}$Acc.⬆ | Acc.⬆ | $\frac{\pi}{6}$Acc.⬆ |
|---|---|---|---|---|---|---|
| **Nuisance** | **Combined** | | **Context** | | **Weather** | |
| **CDAN 27** | .760 | - | .710 | - | .745 | - |
| **BSP 5** | .753 | - | .610 | - | .730 | - |
| **MDD 66** | .780 | - | .761 | - | .802 | - |
| **MCD 42** | .772 | - | .798 | - | .810 | - |
| **MCC 16** | .785 | - | .730 | - | .767 | - |
| **FixBi 33** | .821 | - | .802 | - | .755 | - |
| **MIC 13** | .837 | - | .755 | - | .817 | - |
| **ToAlign 56** | .761 | - | .712 | - | .720 | - |
| **CST 26** | .840 | - | .687 | - | .813 | - |
| **DUA 32** | .699 | - | .667 | - | .701 | - |
| **DINE 24** | .835 | - | .867 | - | .798 | - |
| **DMNT 52** | .811 | .495 | .798 | .524 | .845 | .545 |
| **ORL 15** | .831 | .401 | .848 | .413 | .823 | .389 |
| **Ours (CIDA-3D)** | **.922** | **.556** | **.931** | **.601** | **.901** | **.557** |
| **Nuisance** | **Shape** | | **Pose** | | **Texture** | |
| **CDAN 27** | .820 | - | .844 | - | .773 | - |
| **BSP 5** | .696 | - | .831 | - | .757 | - |
| **MDD 66** | .895 | - | .870 | - | .836 | - |
| **MCD 42** | .896 | - | .865 | - | .834 | - |
| **MCC 16** | .874 | - | .867 | - | .818 | - |
| **FixBi 33** | .854 | - | .842 | - | .801 | - |
| **MIC 13** | .821 | - | .799 | - | .807 | - |
| **ToAlign 56** | .594 | - | .788 | - | .719 | - |
| **CST 26** | .858 | - | .887 | - | .831 | - |
| **DUA 32** | **.918** | - | .755 | - | .695 | - |
| **DINE 24** | .911 | - | .885 | - | .838 | - |
| **DMNT 52** | .796 | .515 | .818 | .380 | .756 | .568 |
| **ORL 15** | .821 | .440 | .869 | .335 | .829 | .439 |
| **Ours (CIDA-3D)** | .910 | **.611** | **.921** | **.459** | **.935** | **.605** |

els to prevent memorization. Our method significantly outperforms state-of-the-art classification UDA methodsRusak et al. (2021); Schneider et al. (2020).

**Synthetic→OOD-CV**    Table 3 show the results on our novel Unsupervised Domain Adaptation setup where we adapt from a synthetic source domain to nuisance-ridden real world data (OOD-CVZhao et al. (2023)). This is a challenging setup which shows that our method is able to bridge the synthetic-real domain gap significantly and we can transfer 3D object pose knowledge learned from synthetic data where it is trivial to generate 3D object pose to real-world nuisance ridden image. This real-world 3D information can be further utilized to robustify downstream computer vision tasks.

Further **experimental and ablation analysis** is deferred to the appendix due to limited space.

## 5  CONCLUSION

In this work, we attempt to solve the problem of unsupervised domain adaptation for 3D-Aware classification and multiclass pose estimation. We focus our efforts on real world data with nuisances like weather, shape, texture, etc. and show that our method is capable of adapting to a nuisance-ridden domain with only unlabeled (and synthetic) image data. Our method further offers the potential to generate 3D pose information for existing real-world image datasets. By training solely on synthetic

Table 2: UDA results for Pascal3d+ → Corrupted-Occluded-Pascal3D+       (Metrics : Classification Accuracy (Acc.), $\pi\backslash 6$ ($\frac{\pi}{6}$) and $\pi\backslash 18$ Accuracy ($\frac{\pi}{18}$))

| Occlusion | F0G0 (0%) | | | F1G1 (20-40%) | | | F2G2 (40-60%) | | |
|---|---|---|---|---|---|---|---|---|---|
| Metric | Acc. | $\frac{\pi}{6}$⬆ | $\frac{\pi}{18}$⬆ | Acc. | $\frac{\pi}{6}$⬆ | $\frac{\pi}{18}$⬆ | Acc. | $\frac{\pi}{6}$⬆ | $\frac{\pi}{18}$⬆ |
| **Spatter Noise** | | | | | | | | | |
| **RPL41** | .749 | - | - | .449 | - | - | .254 | - | - |
| **BNA44** | .693 | - | - | .467 | - | - | .271 | - | - |
| **ORL15** | .815 | .617 | .366 | .685 | .438 | .204 | .484 | .266 | .097 |
| **Ours** | .999 | .825 | .649 | .963 | .594 | .277 | .848 | .424 | .137 |
| **Motion Blur** | | | | | | | | | |
| **RPL41** | .766 | - | - | .545 | - | - | .421 | - | - |
| **BNA44** | .749 | - | - | .556 | - | - | .411 | - | - |
| **ORL15** | .793 | .543 | .284 | .573 | .328 | .122 | .378 | .182 | .054 |
| **Ours** | .996 | .731 | .430 | .956 | .522 | .207 | .822 | .330 | .100 |
| **Snow** | | | | | | | | | |
| **RPL41** | .752 | - | - | .499 | - | - | .389 | - | - |
| **BNA44** | .711 | - | - | .512 | - | - | .469 | - | - |
| **ORL15** | .857 | .565 | .311 | .697 | .410 | .159 | .504 | .215 | .074 |
| **Ours** | .991 | .784 | .493 | .951 | .586 | .271 | .824 | .417 | .145 |
| **Pixelate** | | | | | | | | | |
| **RPL41** | .844 | - | - | .526 | - | - | .331 | - | - |
| **BNA44** | .840 | - | - | .558 | - | - | .395 | - | - |
| **ORL15** | .743 | .444 | .205 | .565 | .273 | .088 | .389 | .152 | .038 |
| **Ours** | .993 | .767 | .486 | .958 | .342 | .159 | .812 | .21 | .101 |
| **Elastic Transform** | | | | | | | | | |
| **RPL41** | .751 | - | - | .455 | - | - | .255 | - | - |
| **BNA44** | .699 | - | - | .471 | - | - | .268 | - | - |
| **ORL15** | .813 | .614 | .371 | .537 | .350 | .160 | .315 | .183 | .068 |
| **Ours** | .994 | .718 | .499 | .972 | .455 | .201 | .878 | .275 | .090 |
| **Shot Noise** | | | | | | | | | |
| **RPL41** | .783 | - | - | .512 | - | - | .119 | - | - |
| **BNA44** | .768 | - | - | .523 | - | - | .243 | - | - |
| **ORL15** | .521 | .323 | .127 | .397 | .156 | .048 | .275 | .092 | .021 |
| **Ours** | .986 | .805 | .534 | .938 | .562 | .253 | .798 | .400 | .152 |

Table 3: Unsupervised Domain Adaptation from Synthetic Data to OODCV 67

| | Acc.⬆ | $\frac{\pi}{6}$Acc.⬆ | Acc.⬆ | $\frac{\pi}{6}$Acc.⬆ | Acc.⬆ | $\frac{\pi}{6}$Acc.⬆ |
|---|---|---|---|---|---|---|
| **Nuisance** | **Combined** | | **Context** | | **Weather** | |
| **CDAN 27** | .650 | - | .609 | - | .653 | - |
| **DUA 32** | .549 | - | .537 | - | .631 | - |
| **DINE 24** | .715 | - | .791 | - | .693 | - |
| **ORL 15** | .803 | .377 | .798 | .396 | .798 | .355 |
| **Ours (CIDA-3D)** | **.902** | **.515** | **.923** | **.591** | **.900** | **.537** |
| **Nuisance** | **Shape** | | **Pose** | | **Texture** | |
| **CDAN 27** | .750 | - | .711 | - | .536 | - |
| **DUA 32** | .811 | - | .677 | - | .544 | - |
| **DINE 24** | .799 | - | .783 | - | .819 | - |
| **ORL 15** | .699 | .410 | .799 | .295 | .791 | .402 |
| **Ours (CIDA-3D)** | **.901** | **.591** | **.920** | **.448** | **.911** | **.601** |

data and validating with human evaluation, this approach could pave the way for enriching common image datasets with corresponding 3D pose annotations.

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

## A    Appendix

## B    Source Model

Our source model is similar to a recently proposed concurrent work Jesslen et al. (2023). Figure 3 shows the inference pipeline for our source model. This model itself is based on a line of work using feature-level neural mesh models and render and compareWang et al. (2021a); Ma et al. (2022); Wang et al. (2023). The difference is that most of the previous work is in single-category versions, whereas our source model trains multiple categories on a single neural backbone. This entails running the contrastive learning training methodology over all mesh vertex features for all classes instead of just one. In addition to this modification, the geometry-independent feature matching is only used in our source model. As noted in the main draft, the source model modifications is not the contribution of our paper and our contributions lie in fully unsupervised adaptation of the source model for both image classification (3D aware classification) and 3D pose estimation.

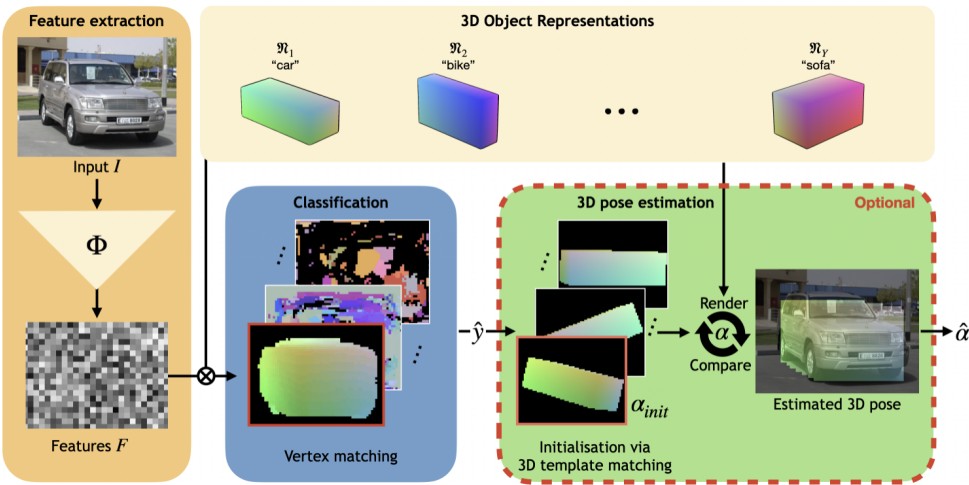

Figure 3: Our source model's inference pipeline. The figure is taken from Jesslen et al. (2023). For geometry-independent feature matching classification, the neural mesh vertex features are utilized without considering their relative positions on the cuboid neural mesh. The objective is to find the maximum number of vertices which are activated for a class given an image's feature map obtained from the neural backbone. Subsequently, the predicted mesh model can be chosen from the classification prediction to run render-and-compare methodology to estimate pose.

## C    Ablation Analysis

**Divergence of Inference results using Geometry-Independent Feature Matching and Render-and-Compare in OOD scenarios**    Table 4 gives classification results for our source model Jesslen et al. (2023) when evaluated on a subset of OOD-CV context nuisance data Zhao et al. (2023) using Geometry-Independent Feature Matching (labeled feature matching) and Render-and-Compare

(labeled pose error). For classification using render-and-compare, we do feature-level render-and-compare for all the categories using individual neural mesh models. Since this process is computationally expensive, we only do it on a subset of the dataset. In our experiments, we find that upto $\sim 20\%$ of samples could be predicted differently by these two classification inference methodologies.

Table 4: Source Model Inference on a OOD-CVZhao et al. (2023) context nuisance data subset

| | Classification Accuracy ⬆ | $\frac{\pi}{6}$ Accuracy ⬆ | $\frac{\pi}{18}$ Accuracy ⬆ |
|---|---|---|---|
| **Feature Matching** | .852 | .429 | .159 |
| **Pose Error** | .794 | .413 | .141 |

**Local Part Robustness in Multi-Class Setting**  Figure 4 shows the visualization of the Before and After CIDA-3D adaptation of robustly detected vertices for a specific category. Figures are for azimuth angles only for a simpler representation. As can be seen, we can still detect robust vertices in a multiclass setting where mesh vertices for each object category are trained using a single backbone. The figures belong to experiments done on the Corrupted-Occluded-Pascal3D+ benchmark and show that the local part robustness hypothesis Kaushik et al. (2024) also holds in a multiclass setting. The post-adaptation subfigure also shows that our method, CIDA-3D, is able to robustly and successfully update the mesh models and backbones in an unsupervised manner to a nuisance-ridden target domain.

**Top-1 and all class vertex update**  Table 5 shows that using just the top-1 prediction from our classification model leads to relatively slower convergence as compared to using our method. Using all class predictions for model update requires pose estimation for classes which is about 5x times slower for the Corrupted-Pascal3D+ experiment on a RTX 2080 GPU. The ablation results shown are from level 5 spatter noise experiment for no occlusion with Corrupted Pascal3D+ benchmark.

Table 5: Source Model Inference on a OOD-CVZhao et al. (2023) context nuisance data subset

| | Average Adaptation Epochs | Classification Accuracy | $\frac{\pi}{6}$ Accuracy |
|---|---|---|---|
| **Top-1** | 200 | .978 | .765 |
| **All** | 58 | .975 | .677 |
| **Ours** | 40 | .999 | .825 |

## D  EXPERIMENTAL DETAILS

For RPLRusak et al. (2021) and BNASchneider et al. (2020), we used the official implementationa. For MCC Jin et al. (2019), CDAN Long et al. (2018), MCD Saito et al. (2018), MDD Zhang et al. (2019) and BSP Chen et al. (2019), we use the Transfer Learning libraryJunguang et al. (2020) implementations. We use the recommended hyperparameters for each method. We utilize a pretrained Imagenet-50 backbone wherever necessary.

## E  LIMITATIONS

Our model shares our limitations with our source model. While the simple cuboid model representation is sufficient for rigid objects, future work involving deformable entities would require more complex mesh modeling. Having multiple neural meshes without shared vertices scales poorly for large number of classes, and a sub-linear neural mesh scaling would be preferred. As the number of categories increases, the complexity of contrastive loss optimization also increases. You may include other additional sections here.

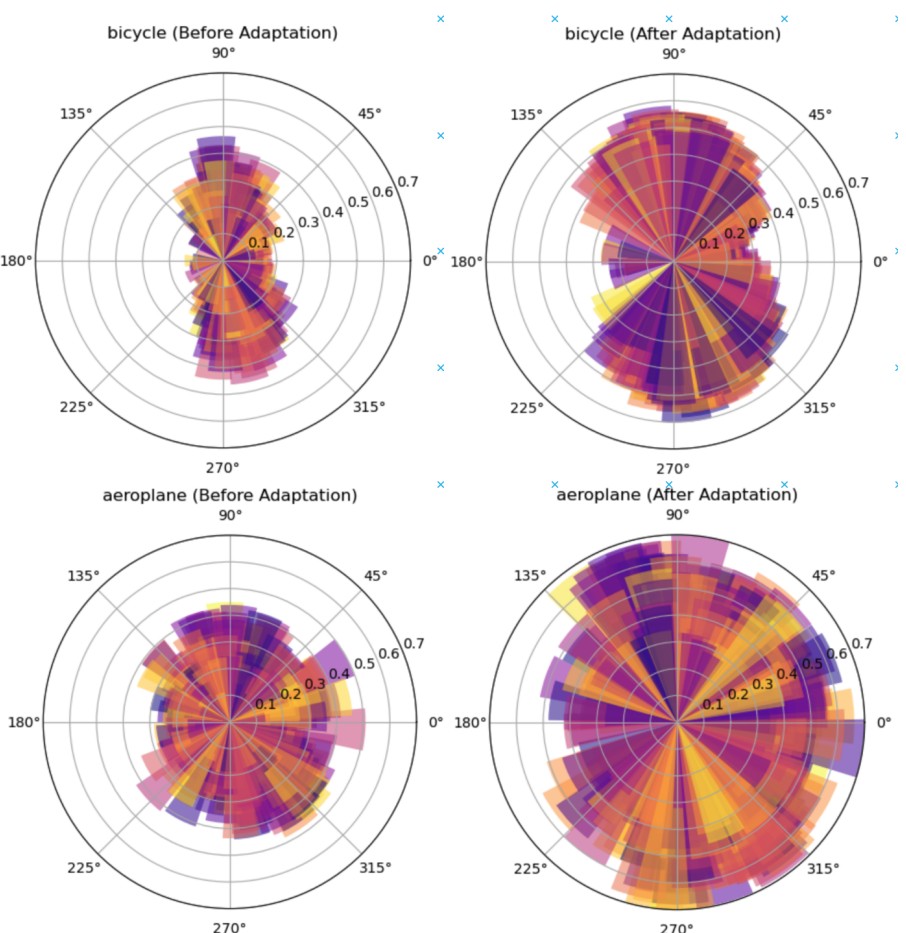

Figure 4: Azimuth Polar histograms representing the ratio of visible neural mesh vertices which are robustly detected for different categories of the Corrupted-Occluded-Pascal3D+ benchmark (for spatter (bicycle) and snow (aeroplane) noise) before and after adaptation using our method. We can see the ratio of robustly detected vertices in the corrupted target domain using the source model which provides evidence towards our hypothesis regarding locally robust neural vertex features in a multi-class setting, similar to Kaushik et al. (2024). Our method, CIDA-3D, like Kaushik et al. (2024) leverages these locally robust parts and adapts the model in an unsupervised manner. The right column shows the increase in ratio of robustly detected vertex features post adaptation using CIDA-3D.

