# OpenReview forum: "CIDA3D: Conformal Inference aided unsupervised Domain Adaptation for 3D-Aware Classification"
_ICLR.cc/2025/Conference — ICLR 2025 Conference Withdrawn Submission_

### Official Review · Reviewer_tEUQ · 2024-11-03

**Soundness:** 2
**Presentation:** 1
**Contribution:** 1
**Rating:** 3
**Confidence:** 3

**Summary:**

This paper aims to address unsupervised domain adaptation for 3D-aware classification and multiclass pose estimation. Based on existing works, 3DUDA and Jesslen et al.(2023), this paper introduces additional conformal prediction to achieve confident prediction sets, and apply it to multi-category UDA setup. Experiments on OOD-CV and two syn-to-real settings show the proposed method outperforms existing methods.

**Strengths:**

The proposed method outperforms existing methods in a large margin as shown in Tabs 1-3.

**Weaknesses:**

1. The overall writing should be improved. The paper is hard to follow.

- Citations are incorrectly formatted throughout the paper. The usage of \citep and \cite should be taken care of. For example, the citations in L.32,33,154 are incorrect.
- The captions of Figs.1 and 2 are too small. Also, the title and the descriptions in Fig.2 (b) are not distinguishable.
- Some of the punctuation marks are incorrect. For example, Eqs.2,4,5, L.361.
- The description of conformal inference in the Related Work is unclear and not connected to the proposed method.
- The notation/formulation should be defined in L.158, however, more notations are re-defined in L.287-307. This makes the formulation hard to follow.
- There are typos in L.427.

2. There is a lack of coherence in the Introduction. This makes it difficult for the reader to understand the motivation. For example, L.31-37 introduces 3D representation and cognitive science independently without conjunctions. The logical connection between sentences needs to be noticed. Moreover, the Introduction only highlights the general challenges of 3D aware classification and domain adaptation but does not provide specific motivation to introduce the proposed framework (i.e.,  local part plurality and robustness, conformal prediction). Last, there is a limited description of the main contributions and their details. This makes the Introduction incomplete and less motivated.

3. L.264 claims that "hypothesis also stands in a multi-class settings". But it is not intuitive based on Fig.1. Fig.1 finds some key observations (L.65-67 "This has been verified... in the multi-class setting" ) without a clear support or evidence. It would be better to introduce some statistical or visual evidence.

4. The novelty of this work is not clear. This work is highly based on Kaushik et al.(2024) and Jesslen et al.(2023). Even though this paper claims it is the first work (L.93), as described in L.85, it seems to only extend existing methods to multi-category UDA setup. I am not sure of the difficulty and the contribution to this extension. Compared Fig.2 in this paper with Fig.2 in Kaushik et al.(2024), the only difference is conformal prediction based on Tibshirani et al. (2019). It would be better to highlight the contribution clearly.

5. It would be better to show the intermediate results (e.g., intermediate feature map visualisation,  intermediate 3D matching) during the optimisation. It is important to see the trend of results (classification and pose) change. Moreover, it would be better to compare the efficiency like training time of different methods. Last, it would be better to show some qualitative results in the main paper or in the supplementary like Fig.3 in Kaushik et al.(2024).

**Questions:**

See Weaknesses

---

### Official Review · Reviewer_8g2v · 2024-11-03

**Soundness:** 2
**Presentation:** 2
**Contribution:** 2
**Rating:** 3
**Confidence:** 2

**Summary:**

This paper presents CIDA-3D, a method that combines Conformal Inference with unsupervised Domain Adaptation for 3D-aware classification and multi-class 3D pose estimation. CIDA-3D employs geometry-independent feature matching for classification and a render-and-compare strategy using neural mesh models for pose estimation. For domain adaptation, CIDA-3D extends the previous work of UDA3D to a multi-class setting by incorporating weighted Conformal Prediction for covariate shift. Experiments are conducted to evaluate the effectiveness of CIDA-3D in adapting from synthetic to real domains.

**Strengths:**

- CIDA-3D extends UDA3D to a multi-class setting, enabling unsupervised domain adaptation for 3D-aware classification and 3D pose estimation, while introducing weighted Conformal Prediction for covariate shift.
- Experiments are conducted across various source and target domains to assess the domain adaptation capabilities of CIDA-3D.

**Weaknesses:**

- This paper contains sections that are highly similar to the content in the 3DUDA paper [1]. For instance, Figures 1 and 2 in both papers are notably alike, and the text in Lines 158-186 closely mirrors the notation and Section 3.1 in the 3DUDA paper.

- The first paragraph in the methodology section places excessive emphasis on the source model and could be relocated to Section 3.1 or revised to serve as an overview of CIDA-3D. Sections 3.1 and 3.2 are not central contributions of the paper and could be further streamlined for conciseness.

- Compared to UDA3D, CIDA-3D operates in a multi-class setting, incorporating Conformal Prediction and addressing local part plurality. However, experimental comparisons are lacking. For instance, what are the pose estimation results when a known category is provided to CIDA-3D, or when a category estimated by CIDA-3D is given to UDA3D?



[1] Prakhar Kaushik et al., 'Source-Free and Image-Only Unsupervised Domain Adaptation for Category Level Object Pose Estimation'.

**Questions:**

See Weaknesses.

---

### Official Review · Reviewer_cWBM · 2024-11-05

**Soundness:** 3
**Presentation:** 2
**Contribution:** 3
**Rating:** 5
**Confidence:** 3

**Summary:**

The authors build on the idea that human perception is robust to occlusions and distribution shifts due to internal 3D representations, a concept that has been used in computer vision to enhance 3D object understanding. CIDA-3D extends this idea by incorporating conformal prediction to handle uncertainty in the adaptation process, enabling better model robustness and efficient adaptation to challenging real-world target domains.

Key contributions of the paper include:

1.The introduction of CIDA-3D, the first method to perform unsupervised domain adaptation for 3D-aware classification and multiclass 3D pose estimation, relying solely on images without labels or 3D information from the target domain.

2.The utilization of local part plurality and local part robustness, leveraging parts of objects that are less affected by out-of-distribution (OOD) inputs to enhance the model’s adaptation capabilities.

3.A novel application of weighted conformal prediction to minimize computational overhead while maintaining confident and reliable prediction sets in the presence of covariate shift.

4.Comprehensive experiments that demonstrate the model’s ability to adapt from synthetic data to complex real-world domains, showing improvements in both classification and 3D pose estimation tasks across several benchmarks with nuisances such as shape, texture, and occlusion.

**Strengths:**

1.The paper addresses a challenging and relatively unexplored problem: adapting 3D-aware models from synthetic data to real-world domains without any labels in the target domain. While previous works focus on supervised or semi-supervised methods for 3D pose estimation and classification, this paper is the first to propose a fully unsupervised approach that performs 3D-aware domain adaptation.

2.The creative combination of conformal prediction with neural mesh models for handling uncertainty in unsupervised domain adaptation is highly original. The use of conformal inference to generate confidence sets in multi-class classification and pose estimation tasks, combined with local part robustness and plurality, represents a novel solution to adapting models to noisy, unlabeled target domains.

3.The paper addresses a highly relevant problem in the field of computer vision, particularly in 3D object recognition and pose estimation under real-world conditions. Given the importance of 3D data in applications like robotics, autonomous driving, and augmented reality, a method that can adapt 3D models from synthetic data to real-world domains without the need for labeled data is highly significant.

**Weaknesses:**

1.CIDA-3D demonstrates strong performance across the three experimental setups presented in this paper, as shown in Tables 1, 2, and 3. The paper also compares its results with several existing methods. The authors claim that this method is the first to attempt unsupervised domain adaptation for robust 3D-aware classification and multiclass pose estimation in real-world scenarios by adapting models trained on procedurally generated synthetic data. I believe that in the context of UDA, adding comparisons with source only and Oracle baselines would provide readers with a better understanding of the performance upper and lower bounds for this task. This would also help future works better assess and benchmark their own methods.

2.I am not an expert in this field, so I found this paper somewhat confusing. It seems that the authors did not organize the overall logic of the paper very well. While the paper introduces the use of conformal prediction to handle uncertainty in domain adaptation, the theoretical justification for why conformal prediction is particularly well-suited for this setting is underdeveloped. The paper explains how conformal prediction generates confidence sets but does not explore why or how this technique is superior to other uncertainty quantification methods (e.g., Bayesian approaches or dropout-based uncertainty).

3.Although the paper is mostly clear, there are occasional clarity issues in how notation is introduced and used, particularly in the Notation section. For instance, key terms like OOD inputs are not fully expanded or explained when first mentioned. This may cause confusion for readers who are not intimately familiar with domain adaptation or 3D object pose estimation.

**Questions:**

1. The paper claims that CIDA-3D is the first to attempt unsupervised domain adaptation (UDA) for 3D-aware classification and pose estimation using synthetic data. However, there are no source-only or Oracle baselines included in the experimental comparisons. Why were these baseline comparisons omitted?

2.While the use of conformal prediction is an interesting choice for handling uncertainty, the paper does not sufficiently explain why conformal prediction is particularly well-suited for the UDA setting. Could the authors provide more details or justification for why this technique was chosen over other uncertainty quantification methods, such as Bayesian approaches or dropout-based methods?

3.I found the overall organization of the paper somewhat confusing, especially regarding the flow of concepts and the introduction of key terms. For example, important terms like OOD inputs and are not fully expanded or explained when first mentioned, which could be confusing for readers who are less familiar with these terms. Could the authors revise these sections for better clarity? Both Table 1 and Table 3 have highlighted the best-performing metrics, but Table 2 does not. Was this an oversight?

---

### Official Review · Reviewer_k5Un · 2024-11-06

**Soundness:** 2
**Presentation:** 2
**Contribution:** 2
**Rating:** 5
**Confidence:** 2

**Summary:**

This paper studies 3D-Aware Classification and multiclass 3D pose estimation. It introduces a new framework CIDA-3D, which leverages unsupervised domain adaptation into these two tasks. The proposed method doesn't requires  3D data or object labels and only use unlabeled image in the target domain, allowing for generalized feature updates across multiple classes, even in noisy target domains. Via experiments, the proposed method is demonstrated to be able to adapt from synthetic to complex real-world target domains.

**Strengths:**

The strengths of this work can be summarized as below:
- CIDA-3D introduces unsupervised domain adaptation into the tasks of 3D-aware classification and multiclass 3D pose estimation.
- In target domain, 3D data or object labels are needed, and only unlabeled images are used.
- The local part plurality and robustness hypothesis is well motivated and sound. The model builds on previous method 3DUDA by utilizing local part plurality and robustness to enhance the adaptation process, allowing the transfer of features across different classes in noisy or occluded environments.
- CIDA-3D uses conformal prediction to manage covariate shifts between source and target domains, which reduces computational overhead while maintaining confidence in predictions.
- The experiments demonstrate that CIDA-3D enables synthetic-to-real domain adaptation, bridging the sim2real domain gap and enhancing robustness against complex real-world nuisances such as texture, shape, and environmental variations.

**Weaknesses:**

The weaknesses of this work are summarized below:
- CIDA-3D relies on a domain classifier to distinguish two domain. This is practical but could introduce dependency on the classifier’s accuracy and increase complexity. The whole system is vulnerable to misclassification, and potentially lead to less effective conformal prediction. It's unclear how robust and what's the capacity of this domain classifier, or if it's possible to take alternative classifier-free UDA solutions.
- The ablation studies lack depth regarding key components like the choice of neural mesh and conformal prediction. A detailed analysis of the important design choices, parameters, alternative solutions could help demonstrate the effectiveness of these models.
- Since this paper tackles classification, to fully establish CIDA-3D’s generalizability it's crucial to test on large-scale classification datasets. For example, test on imagenet or openimage dataset where more object categories are presented. The current scope may limit the reader’s ability to assess performance in diverse applications, which would be especially important for real-world implementations.

**Questions:**

- The local parts robustness is reasonable, but this paper does not explore how different levels of local feature robustness influence performance. For example, many objects may exhibit different robustness levels in different local parts. A more fine-grained analysis could be very insightful.
- Since Conformal Prediction is the main contribution of this work, I'm wondering if it's possible to demonstrate the Interpretability of this module, to help us better understand how well does this module work.
- In L270-282 the authors explains on why conformal prediction was chosen over alternative uncertainty management methods. I'm wondering if there are empirical evidence to support these claims.
- Is it possible to provide a way or metric to evaluate local part robustness and plurality in different dataset?

**Details Of Ethics Concerns:**

The paper template was modified:
- The font size of the figure caption is changed to be smaller
- In all the tables the citations become numbers rather than author names.

I have read the author guidance and cannot determine the severeness of above things. They probably are not a big deal but I raise it here for reference.

---

### Note · Authors · 2024-11-15

**Comment:**

We thank the reviewers for their time and effort in reviewing this work. On closer inspection of the reviews, we have decided to withdraw our paper and further improve it by including a number of suggestions and experiments mentioned by the reviewers.

**Withdrawal Confirmation:**

I have read and agree with the venue's withdrawal policy on behalf of myself and my co-authors.